# Capabilities of Terahertz Cyclotron and Undulator Radiation from Short Ultrarelativistic Electron Bunches

**Vladimir Bratman [1,2,\*], Yuri Lurie [1], Yuliya Oparina [2,\*] and Andrey Savilov [2]**

[1]  Electrical and Electronic Engineering, Ariel University, 40700 Ariel, Israel; ylurie@ariel.ac.il
[2]  Institute of Applied Physics, Russian Academy of Sciences, 603950 Nizhny Novgorod, Russia; savilov@appl.sci-nnov.ru
\*  Correspondence: v_bratman@yahoo.com (V.B.); oparina@appl.sci-nnov.ru (Y.O.)

**Abstract:** Mechanisms of coherent spontaneous cyclotron and undulator radiations of short dense bunches, in which electrons move along the same stationary helical trajectories, but have different dynamic properties, have been compared in detail. The results are based on the simplest 1D model in the form of a plane consisting of uniformly distributed synchronously moving and in-phase emitting particles, as well as numerical 3D codes developed to study the dynamics of bunches in waveguides taking into account the effects of the radiation and spatial charge fields. For cyclotron radiation under group synchronism conditions, the Coulomb expansion of a bunch occurs along the surface of a constant wave phase with the formation of an effectively radiating coherent structure. A significantly higher radiation frequency, but with a lower efficiency, can be obtained in the regime of simultaneous excitation of high-frequency (autoresonant) and low-frequency waves; in the field of the latter, stabilization of the bunch phase size can be achieved. Such a two-wave generation is much more efficient when the bunches radiate in the combined undulator and strong guiding magnetic fields under conditions of the negative mass instability, when both the Coulomb interaction of the particles and the radiation field stabilize the longitudinal size of the bunch.

**Keywords:** THz radiation; free electron lasers; cyclotron and undulator radiation; negative mass instability; autoresonance

## 1. Introduction

Among a large number of ideas put forward in [1] for generating in the submillimeter and shorter wavelength ranges, V. Ginzburg has proposed using Doppler-frequency up-converted cyclotron and undulator radiations of short bunches of ultrarelativistic electrons. For pumping of transverse oscillations of particles, V. Ginzburg suggested an undulator in the form of an alternating electric field, and soon, independently, H. Motz suggested a more convenient magnetic undulator for this purpose and implemented his idea in the experiment [2]. Preshaping dense bunches with a longitudinal size less than the length of the emitted wave and with a sufficiently large charge needed to produce high-power and high-frequency coherent spontaneous radiation (CSR) for a long time was too difficult, although much effort was spent to solve the problem [3]. As with a number of other radiation mechanisms, it turned out to be much easier to create highly efficient generators based on stimulated cyclotron and undulator radiations, in which the particles are self-consistently collected into compact dense bunches under the action of the wave they radiate. Cyclotron radiation masers, and a gyrotron as their most developed variety, make it possible to obtain an unprecedented high radiation power in the millimeter and submillimeter wavelength ranges, while free-electron lasers, which are based on

undulator radiation, generate much shorter waves and already operate in the X-ray range. At the same time, the development of the accelerator technology and, in particular, the emergence of laser-driven photo injectors, which permit one to obtain dense picosecond and sub-picosecond electron bunches with a charge of the order of 1 nC or more, made it possible and attractive again to come back to the original ideas of V. Ginzburg and H. Motz to generate submillimeter (terahertz) radiation. The bunches with a relatively low particle energy of 3–6 MeV produced in photo injectors can be directly used to produce high-power pulsed generation based on various electron emission mechanisms in the terahertz frequency range [4–12].

Two efficient radiation mechanisms are considered and compared in this paper, namely, (1) cyclotron radiation of bunches moving in a uniform magnetic field [13] and (2) undulator radiation of bunches moving in the field of a helical undulator and in a strong (super-resonant) uniform longitudinal guiding field under conditions where the negative mass (NM) instability can stabilize the longitudinal size for a dense bunch core [14,15]. In these two cases, the electrons in the bunch can move near identical stationary helical trajectories, but the properties of these "electron oscillators" and the mechanisms of their emission and bunching/repulsion in the wave field and the Coulomb field of the other particles differ significantly. In Section 2 of this paper, we use for both mechanisms the model of an ideal one-dimensional CSR source in the form of a uniformly charged plane composed of in-phase emitting electrons moving along identical trajectories in the corresponding fields. This model permits one to find the limiting values of efficiency and reveals important features of the mechanisms considered. Three-dimensional simulation and comparison of more complex effects associated with the finite longitudinal size of the bunches moving in cylindrical waveguides, the Coulomb interaction of particles inside the bunches and the velocity spread, as well as frequency dispersion and the excitation of various waveguide modes, are done in Sections 3 and 4. The results of this work are summarized in Section 5.

## 2. 1D Model of a Perfect CSR Source

Consider the radiation of electrons moving along helical trajectories in two types of magnetic fields, namely, (1) in a uniform field $\boldsymbol{H_0} = H_0\boldsymbol{z_0}$ and (2) in the combined field $\boldsymbol{H_u}$ of a helical undulator and a strong (super-resonant) uniform field. In the first case, the particles perform free transverse oscillations at the cyclotron frequency $\omega_c = eH_0/mc\gamma$ and move at a constant translational velocity $v_z$. Here, $e$, $m$, and $\gamma$ are the electron's charge, mass, and Lorentz factor, respectively, and $c$ is the speed of light. In the second case, we consider the case where the particles also move along helical trajectories, but oscillate with the frequency of forced undulator oscillations $\omega_u = 2\pi v_z/d$ and cyclotron oscillations are not excited: $d$ is the undulator period and $v_z$ is the constant longitudinal velocity. In the second case, the transverse particle velocity normalized to the speed of light is determined by the expression (see e.g. [16] and Equations (10) and (11))

$$\boldsymbol{\beta_\perp} = -\left(\frac{K}{\Delta}\right)\left(\boldsymbol{x_0}cos\omega_u t + \boldsymbol{y_0}sin\omega_u t\right), \tag{1}$$

where $K = \frac{eH_u d}{2\pi mc^2}$ is the undulator parameter and $\Delta = 1 - \frac{\omega_c}{\omega_u}$ is the relative mismatch of the resonance between free and forced oscillations. For the further analysis, it is important to note that if the energy and velocity of a particle change under conditions where it deviates little from the stationary trajectory, the derivative

$$\frac{d\beta_z}{d\gamma} \approx \frac{1}{\gamma^3}\left(1 + \frac{K^2}{\Delta^3}\right) \tag{2}$$

can be negative in a strong (super-resonance) guiding magnetic field if the resonance mismatch is small enough [15]:

$$\Delta < 0, \ |\Delta|^3 < K^2 \tag{3}$$

Under these conditions, an increase (decrease) in the particle energy in the Coulomb field of other particles, as well as in the field of the emitted waves, can lead to a decrease (increase) in its translational velocity. Such an "anomalous" behavior of a particle is associated with a rapid increase (decrease) in its transverse velocity near the resonance. A consequence of this change in the longitudinal velocity can be effective mutual attraction of particles during their Coulomb interaction, which stabilizes the bunch in the longitudinal direction (the NM effect in the undulator), as well as bunching of the particles in the wave field.

An electron moving along a helical trajectory radiates at frequencies $\omega$ satisfying the Doppler-shifted resonance condition with the emitted wave at any harmonics of the oscillation frequency $\Omega$:

$$\omega - k_z v_z = s\Omega, \;\; s = 1, 2, 3 \dots. \tag{4}$$

here, $\Omega = \omega_c$ for cyclotron and $\Omega = \omega_u$ for undulator radiation; $k_z$ is the longitudinal wave number.

The dynamics of a radiating electron bunch with a large charge is determined by the radiation field excited by it, as well as by the strong field of its own space charge. Before considering the whole complex of effects associated with the radiation of a short dense bunch in a waveguide, let us consider the radiation of a simple source, the important features of which with some modifications appear in more complex problems. Namely, we consider radiation in free space of a perfect 1D CSR source in the form of a moving plane, perpendicular to a uniform magnetic field and formed by uniformly distributed electrons, which synchronously move in this field along helical trajectories and emit Doppler-converted plane waves in phase at the fundamental harmonic of oscillations in the positive and negative directions of the z axis (Figure 1).

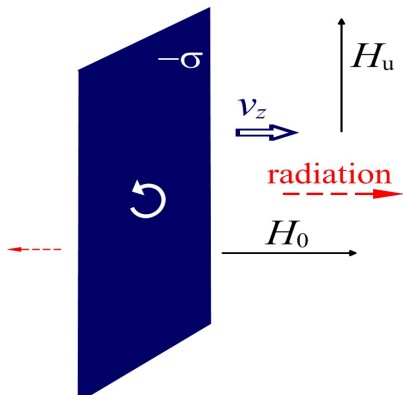

**Figure 1.** Perfect 1D source of Coherent Spontaneous Radiation (CSR) in the form of a moving plane formed by uniformly distributed electrons, which synchronously move along identical helical trajectories and emit plane waves in the positive and negative directions of the z axis.

The transverse and longitudinal fields generated by such a plane with a surface electric current $-\sigma v_\perp(t)$ in the cross-section z are defined by the expressions [17]

$$\boldsymbol{E}_\perp = 2\pi\sigma \frac{\boldsymbol{\beta}_\perp(\tilde{t})}{\left[1 \mp \beta_z(\tilde{t})\right]}, \boldsymbol{H}_\perp = \pm \boldsymbol{z_0} \times \boldsymbol{E}_\perp, E_z = \mp 2\pi\sigma. \tag{5}$$

Here, $-\sigma$ is the surface charge density, $\tilde{t} = t - \frac{|z - z_0(t)|}{c}$ is the retarded time, $z_0(t)$ is a longitudinal coordinate of the radiating particle and the entire plane as a whole at time $t$. The upper and lower signs in (5) correspond to the front and rear half-spaces with respect to the moving plane, respectively.

The transverse fields affecting the electrons are equal to the arithmetic averages of the fields given by Equation (5) on both sides of the plane:

$$E'_\perp = 2\pi\sigma \frac{\boldsymbol{\beta}_\perp(t)}{\left[1 - \beta_z^2(t)\right]}, \; H'_\perp = \beta_z \cdot \boldsymbol{z_0} \times E'_\perp. \tag{6}$$

For cyclotron radiation of the emitting plane, the equations of motion for a single particle in a uniform magnetic field and the field of the plane (6) have an integral $v_z = const$, the existence of which is evident upon transition to the plane's reference system. From this integral it is clear that the minimum energy of the radiating particle is achieved when the transverse velocity vanishes and is limited to the value of $\gamma_z = 1/\sqrt{1 - \beta_z^2}$. The complete system of equations of electron motion is reduced to the equation for energy, which is easily integrated and gives an explicit dependence of $t$ on the normalized electron energy $w = \gamma/\gamma_z$ (cf. [17]):

$$w - w_0 + \frac{1}{2}\left(ln\frac{w - 1}{w + 1} - ln\frac{w_0 - 1}{w_0 + 1}\right) = -2\delta t. \tag{7}$$

Here, $w_0 = \gamma_0/\gamma_z$, $\gamma_0$ is the initial particle Lorentz factor and $\delta = e\pi\sigma/mc$. At the start of the CSR, the synchronously moving particles of the plane lose energy according to a law close to a linear dependence on time, and then they approach the lowest energy asymptotically in infinite time with a decay rate proportional to $\delta$ (Figure 2a).

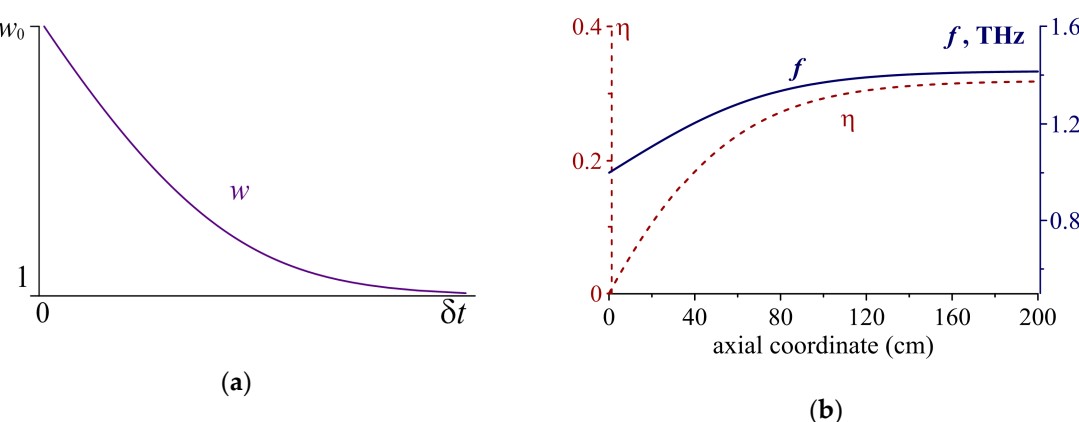

(**a**)

(**b**)

**Figure 2.** Cyclotron radiation of a moving plane: (**a**) change in electron energy; (**b**) current radiation frequency and efficiency for an initial particle energy of 6 MeV, normalized transverse velocity $\beta_{\perp 0} = \frac{1}{\gamma_0}$, surface density $\sigma = 1.2 \times 10^{-4} C/m^2$, and magnetic field $H_0 = 27$ kOe, $H_u = 0$.

With the characteristic value of the initial transverse velocity $\beta_{\perp 0} = \frac{1}{\gamma_0}$ of ultrarelativistic $(\gamma_0 >> 1)$ particles, a significant fraction of the energy is converted into radiation; the efficiency of this process is:

$$\eta = \frac{\gamma_0 - \gamma}{\gamma_0 - 1} \approx 1 - \frac{1}{\sqrt{2}}. \tag{8}$$

For any transverse particle velocity, the moving plane radiates only in the ± z directions at the fundamental cyclotron harmonic $s = 1$. The main fraction of this energy is radiated forward with the instantaneous radiation frequency

$$\omega_+ = \frac{\omega_c}{(1 - \beta_z)} \approx 2\gamma_z^2\omega_c, \tag{9}$$

while the fraction of backward radiation at a low frequency $\omega_- = \omega_c/2$ is small, of the order of $\frac{\eta}{4\gamma_z^2}$. This estimate takes into account that the same number of quanta is emitted into forward and backward waves. At a constant translational velocity of electrons, both radiation frequencies monotonously increase, and quite significantly, due to a change in the cyclotron frequency during the radiation process (Figure 2b; here, $f = \omega_+/2\pi$).

For undulator radiation in the combined field of a helical undulator and a uniform guiding field, the longitudinal velocity of the particles is not maintained. In this case, the particle motion is described by the following system of equations:

$$\frac{dp_+}{d\zeta} = -q\frac{p_+}{p_z} + i(1-\Delta)p_+ - iKe^{-i\zeta}, \quad \frac{dp_z}{d\zeta} = -q\frac{p_\perp^2}{1+p_\perp^2} + K\cdot Im\left(\frac{p_+e^{-i\zeta}}{p_z}\right). \tag{10}$$

here, $p_\perp$ and $p_z$ are the transverse and longitudinal components of the electron momentum normalized by the factor $mc$, to which the particle Lorentz factor is related by the formula $\gamma = \sqrt{1+p_\perp^2+p_z^2}$, $p_+ = p_x + ip_y$ is the complex combination of transverse Cartesian components of the momentum, $\zeta = k_u z$ is a dimensionless longitudinal coordinate, $k_u = \frac{2\pi}{d}$, and the parameter $q = \frac{e\sigma d}{mc^2}$ is proportional to the surface charge of the plane. For $q = 0$, Equation (9) gives the solution corresponding to the motion of a single particle along a stationary helical trajectory:

$$p_+ = p_{\perp 0}e^{i\zeta}, \quad p_{\perp 0} = -\frac{K}{\Delta} = const, \quad p_z = p_{z,0} = const, \tag{11}$$

which is equivalent to Equation (1). In the case of perfect injection of a particle into a stationary trajectory the initial conditions for Equations (10) are represented as $p_+(0) = p_{\perp,0}$ and $p_z(0) = p_{z,0}$. If the parameter $q$ is sufficiently small, the particles of the plane experience a relatively weak radiation reaction and deviate weakly from the stationary trajectory at the start of the CSR. If conditions (3) are fulfilled, then a decrease in the energy during radiation is accompanied by an increase in the longitudinal velocity of the particles and radiation frequency, which is confirmed by the numerical solution of Equation (9) (Figure 3). Then this "anomalous" behavior is replaced by "normal," in which the longitudinal velocity and frequency of radiation decrease in the radiation process.

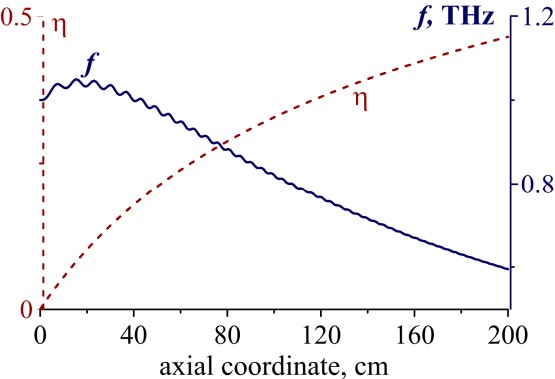

**Figure 3.** Undulator radiation of the moving plane: current radiation frequency and efficiency for an initial particle energy of 6 MeV, transverse velocity $\beta_{\perp 0} = \frac{1}{\gamma_0}$, surface density $\sigma = 1.2\cdot10^{-4}C/m^2$, amplitude of undulator field $H_0 = 40.5$ kOe, and $H_u = 1$ kOe.

## 3. Cyclotron Radiation of an Electron Bunch in a Cylindrical Waveguide

From the radiation of plane waves by a charged plane moving in free space, we proceed to the emission of bunches in a circular waveguide. In this case, the radiation field of a particle can

be represented as a set of normal waves with discrete transverse wave numbers $k_\perp$, for which the longitudinal wave numbers are associated with the emitted frequencies by the dispersion relation

$$\omega^2 = c^2\left(k_\perp^2 + k_z^2\right). \tag{12}$$

The resonance condition between electrons and the radiated monochromatic component of the waveguide mode can be written in the form

$$\omega = \omega_c / \left(1 - \beta_z \beta_{gr}\right) \tag{13}$$

where $\beta_{gr} = \frac{k_z}{k}$ is the normalized group velocity of the wave. In accordance with Equations (12) and (13), there are two characteristic regimes of radiation for electrons moving in a waveguide depending on the value of the magnetic field (Figure 4): (1) intersection of the dispersion characteristics of the beam (Equation (13)) and the wave ( Equation (12)), at which high-frequency (H) and low-frequency (L) waves are simultaneously excited, and (2) tangency of dispersion characteristics, or group synchronism (G), at which the group velocity of the wave coincides with the initial longitudinal velocity of the  electrons.

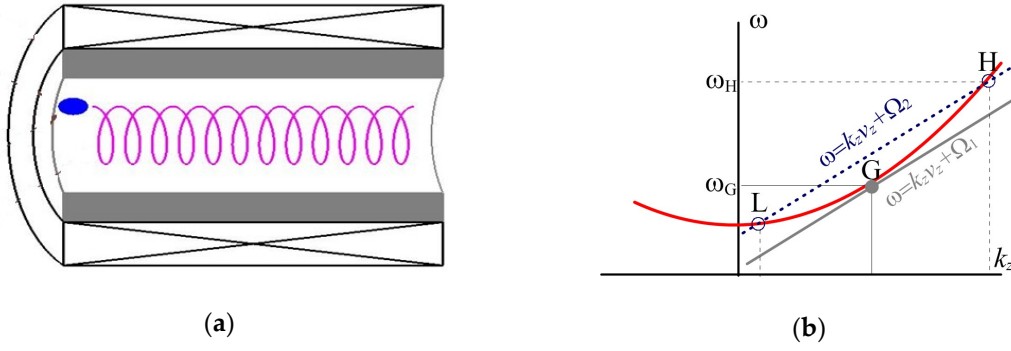

(**a**)                                                    (**b**)

**Figure 4.** (**a**) The cross section of a cylindrical waveguide in which an electron bunch moves along helical trajectories; (**b**) dispersion characteristic for the cases of the electron velocity is close to the group velocity of the radiated wave ("grazing", G), and for the two waves at low (L) and at high (H) frequencies generation regime.

We assume that the initial shape of the bunch is close to cylindrical, with a radius $R_b$ much smaller than the waveguide radius $R_w$ and an axis coinciding with the axis of the waveguide. At the initial time, the particle density distribution in the longitudinal and transverse directions are described by the Gaussian dependences, and in the simplest case the transverse and longitudinal velocities of all particles are considered equal (later, effects associated with the variation of the particle parameters will be considered). In the study of the dynamics of bunches and radiation in this paper, two computational approaches are used. In the approximate method [13,18], the bunch is represented by a set of solid uniformly charged disks whose centers are initially placed on the waveguide axis; the disks interact with the radiation field and between each other, and the radiation field is represented by the lowest $TE_{11}$ wave with a smoothly varying amplitude. In a more general and precise spatial-frequency approach WB3D [19], macroparticles are used, which are thrown at the initial moment into the waveguide and interact with each other and with the radiation field represented by a set of waveguide modes excited at any cyclotron harmonic. The interaction of the particles with each other (which is responsible for a correct description of space-charge forces affecting the bunch) is described by linearized formulas, following from the Lienard–Wichert potentials (see [15] for more details).

Simulations for the simplest case of a solid electron disk formed by ultrarelativistic electrons synchronously moving along identical helical trajectories in a uniform magnetic field along the waveguide axis and radiating at the lowest $TE_{11}$ mode of a circular waveguide at the fundamental cyclotron harmonic show a good agreement with the results for the plane in free space. Namely, the

current frequency and efficiency of radiation for such a disk in the regimes of group synchronism and high-frequency intersection of dispersion characteristics little differ from the corresponding values for radiation of the plane when both radii of the waveguide, $R_{\rm w}$, and the disk, $R_{\rm d}$, are sufficiently large and the ratio of surface densities of the disk and plane (form-factor) is $\frac{\sigma_d}{\sigma} = 0.8\left(\frac{R_w^2}{R_d^2}\right)$, where the coefficient 0.4 is equal to the norm of the wave. It should be mentioned that according to simulations, the model of a solid disk set [13,18] is a fairly accurate approximation for the description of Coulomb interaction inside a bunch and its radiation in a relatively narrow waveguide.

The highest Doppler-frequency up-conversion of the radiation is achieved at the high-frequency intersection of characteristics (point H in Figure 4). In this case, the phase, $v_{\rm ph}$, and group, $v_{\rm gr}$, velocities of the wave are close to the speed of light, the longitudinal velocity of the particles changes, but these changes are small. The case of relatively low magnetic fields, where the lower intersection corresponds not to a backward, but to a forward wave with respect to the particle motion, is most interesting for radiation in the waveguide. Herein, the low frequency is not as different from that of a high-frequency wave as for the backward wave radiated by the plane. Correspondingly, the fraction of radiation into the low-frequency waveguide mode is much larger. The considered high-frequency mode for a bunch is similar in its properties to that used as the operating one for long beams in the so-called cyclotron autoresonance masers (CARMs) [20,21]. It is well known that CARMs are very critical to the spread in particle parameters and to the excitation of low-frequency waves with a small group velocity.

As for the cyclotron superradiation of extended bunches whose lengths are much greater than the radiated wavelength (see, for example, [22,23] and references therein), which is well studied both theoretically and experimentally at millimeter wavelengths, the group synchronism regime provides the highest efficiency in the case of short bunches, whose longitudinal lengths are smaller than the radiated wavelengths. In this case, the radiation field does not slip relative to the beam, but accompanies it, which gives the radiation features of the stimulated process and contributes to the achievement of a high efficiency. At the same time, the radiation frequency in this case is approximately $\frac{\gamma_0^2}{\gamma_{\|0}^2}$ times lower than in the high-frequency mode.

Let us estimate the change in electron parameters during radiation of a dense bunch in the waveguide. The change in the electron energy can be written as (see [13] for more details):

$$\frac{d\gamma}{d(\omega t)} = F_w + F_c, \quad F_w \propto Re\,(E_{+,w}p_+), \quad F_c \propto E_{z,c}. \tag{14}$$

Here, $E_{+,w} = E_{x,w} + iE_{y,w}$ describes the transverse electric field of the wave, $E_{z,c}$ is the axial Coulomb field inside the bunch. For the lowest $TE_{11}$ mode, the transverse fields are related by $H_{+,w} = i\frac{k_z}{k}E_{+,w}$, $\frac{k_z}{k} = \beta_{gr}$. Therefore, the change in the particle axial momentum is

$$\frac{dp_{z,w}}{d(\omega t)} = \beta_{gr}F_w + \beta_z F_c. \tag{15}$$

The change in the energy of an electron in a bunch is the sum of the changes in energy due to the Coulomb interaction and the wave effect: $\Delta\gamma = \Delta\gamma_c + \Delta\gamma_w$. The changes in the normalized longitudinal momentum and the Lorentz factor associated with the Coulomb interaction are correlated as $\Delta p_{z,c} = \Delta\gamma_c/\beta_{z,0}$. The change in the longitudinal particle momentum, which is determined by the interaction with the wave, is related to the change in energy as the transverse components of the electric and magnetic fields of the radiated wave [compare (14) and (15)]: $\Delta p_{z,w} = \beta_{gr}\Delta\gamma_w$. Corresponding changes in the normalized longitudinal velocities are approximately equal to

$$\Delta\beta_{z,c} \approx \frac{1}{\beta_{z,0}\gamma_{z,0}^2}\frac{\Delta\gamma_c}{\gamma_0}, \quad \Delta\beta_{z,w} \approx \left(\beta_{gr} - \beta_{z,0}\right)\frac{\Delta\gamma_w}{\gamma_0}.$$

These formulas allow one to present the inertial part of the change in the resonance phase of the electron in the radiated wave $\vartheta = \omega t - k_z z - \int \omega_c dt$ as

$$\frac{d\vartheta}{d(\omega t)} = \frac{\Delta\gamma_c}{\gamma_0}\left[\left(1 - \beta_{z,0}\beta_{gr}\right) - \frac{\beta_{gr}}{\beta_{z,0}\gamma_{z,0}^2}\right] + \frac{\Delta\gamma_w}{\gamma_0}\left[\left(1 - \beta_{z,0}\beta_{gr}\right) - \beta_{gr}(\beta_z - \beta_{z,0})\right]. \tag{16}$$

According to the approximate Equation (16), the Coulomb interaction of the particles does not change the phase of the electron relative to the wave in the regime of group synchronism, $\beta_{z,0} = \beta_{gr}$, despite the possible expansion of the bunch [13]. At the same time, in the high-frequency regime (H in Figure 4), where $\beta_{gr} > \beta_{z,0}$ and an electromagnetic pulse overtakes the electron bunch, the Coulomb interaction changes the phase of the electron relative to the wave and also weakens the effect of the wave on the particle. In the limit of an exact cyclotron autoresonance regime, where $\beta_{gr} = 1$, the change in cyclotron frequency and the longitudinal velocity of a particle completely compensate for each other [24,25]. Therefore, an additional stabilization of the particle phase size is necessary to maintain the efficiency of the bunch radiation in this regime. It is important that such stabilization can be provided by a low-frequency wave L, which is excited simultaneously with a high-frequency wave.

The resonance phases of particles change differently in the field of low-frequency and high-frequency waves. At the start of the interaction, the phases of the electrons of the bunch moving ahead are smaller than the phases of electrons of the opposite edge of the bunch, and the phase distribution along the length is close to a linear one. According to Equation (16), for interaction with a low-frequency wave, where $\beta_{z,0} > \beta_{gr,L}$, an increase (decrease) in the particle energy due to Coulomb interaction leads to an increase (decrease) in the phase; namely, the phase of the front electrons increases, while the phase of the rear ones decreases. Thus, although the Coulomb fields lead to an increase in the length of the bunch, its phase size even decreases. For the high-frequency wave, the Coulomb field leads to the opposite effect since $\beta_{z,0} > \beta_{gr,H}$. The electron-wave interaction stabilizes the phase size with respect to both waves; however, the center of phasing by the wave shifts to the front of the bunch. Due to a sufficiently large slippage of the low-frequency radiation pulse relative to the bunch (the wave pulse lags behind the particles), the center of the bunch shifts rapidly from the maximum of the decelerating phase towards the stable neutral phase of the wave. In this case, the electron velocity at the edges of the bunch increases more slowly than at the center, which leads to the formation of a peak of the charge density. The high-frequency wave pulse slightly overtakes the electron bunch. For a sufficient shift from the group synchronism regime, where $\left|\beta_{z,0} - \beta_{gr,L}\right| > \left|\beta_{z,0} - \beta_{gr,H}\right|$, the influence of both wave components on the phase change in the high-frequency wave is small. Nevertheless, the phase size of the bunch relative to the high-frequency wave increases rapidly, because at the start of the interaction the Coulomb field in a dense bunch is large and the wave amplitude is small. As a result, Coulomb debunching dominates wave bunching.

As an example, consider the radiation of a cylindrical bunch with a diameter of 1 mm, a charge of 0.1 nC, a duration of 0.25 ps, a particle energy of 6 MeV and initial transverse velocities $\beta_{\perp 0} = \frac{1}{\gamma_0}$ in a waveguide with a diameter of 4 mm. The particles are in group synchronism with the $TE_{11}$ mode at the field $H_0$ = 22 kOe and emit in this case a broadband pulse with a center frequency of about 0.4 THz and a relatively high efficiency of about 10% (Figure 5).

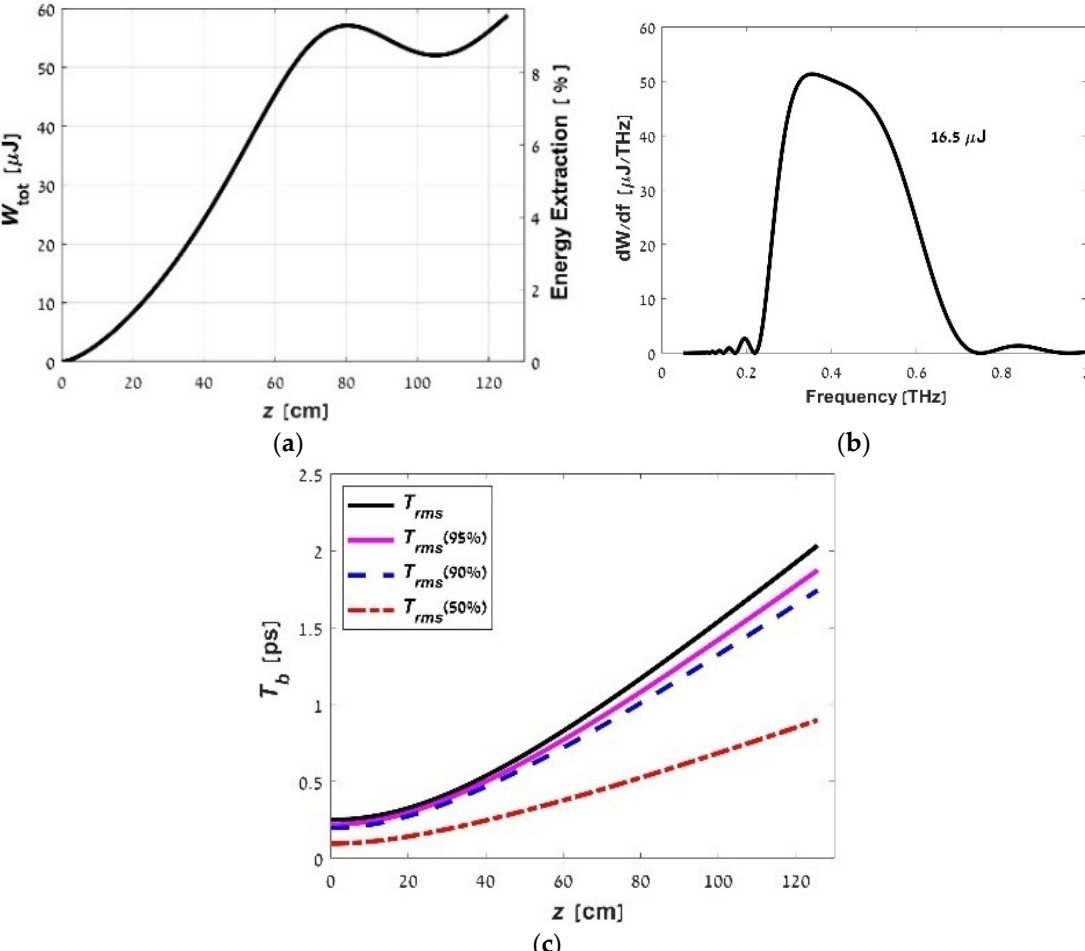

**Figure 5.** Cyclotron CSR of an electron bunch with a diameter of 1 mm, an initial duration of 0.25 ps, a charge of 0.1 nC, and a particle energy of 6 MeV in the group synchronism regime with the $TE_{11}$ mode: (**a**) radiated energy, (**b**) emission spectrum, and (**c**) change in the rms durations for all the bunch and for its central part (without a long tail of particles).

With another field value of 40 kOe, the bunch emits in the same waveguide simultaneously at frequencies of about 0.12 THz and 1.35 THz. The efficiency of high-frequency (autoresonance) radiation with a charge of 0.5 nC higher than in the previous example is close to only 2% (Figure 6). The relatively low efficiency is due to the fact that in this case the Coulomb repulsion of particles exceeds the weaker effect of cyclotron compression by the radiation field [26]. It is important that because of the relatively rapid saturation of radiation energy, the initial spread of the particle pitch-angles ~$1/\gamma$, which corresponds to the longitudinal velocity spread ~0.1%, has little effect on the efficiency.

Generation at high frequencies would be more efficient if the radiation and the Coulomb fields work in the same direction, providing the electron bunch stabilization. This is possible in the NM regime in an undulator with a super-resonance guiding magnetic field, which will be discussed in the next section.

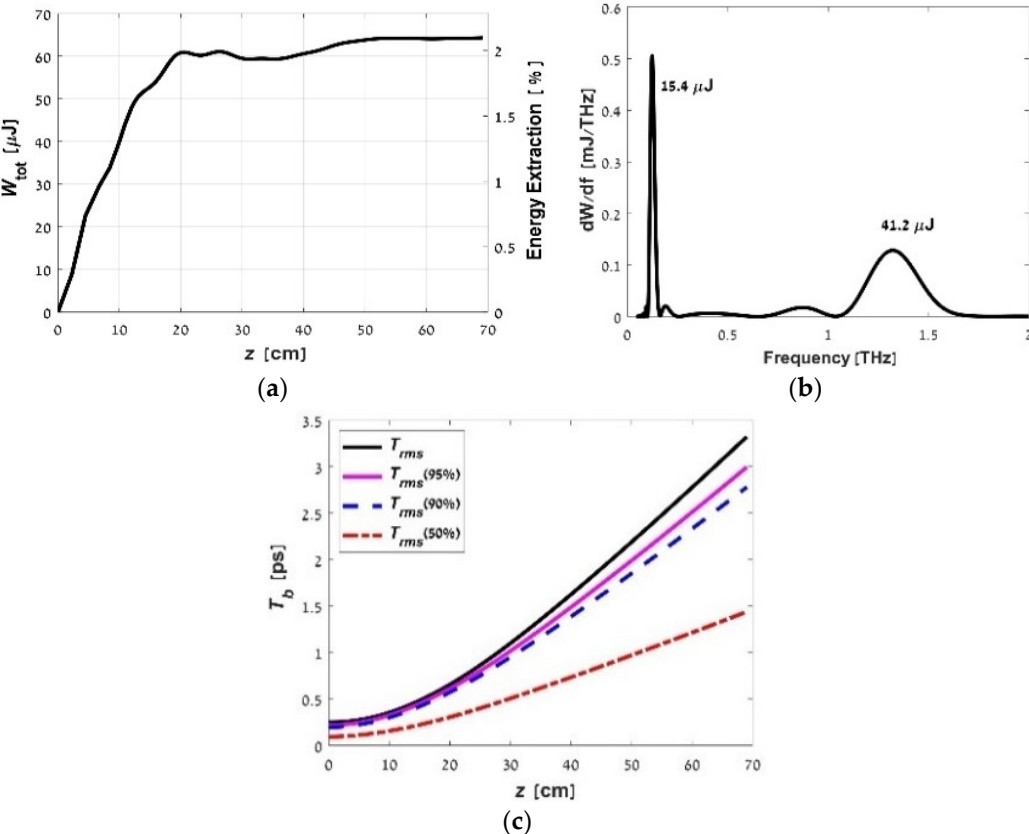

**Figure 6.** Cyclotron CSR of a bunch with a diameter of 1 mm, an initial duration of 0.25 ps, a charge of 0.5 nC, and a particle energy of 6 MeV in the regime of simultaneous excitation of low- and high-frequency waves: (**a**) radiated energy, (**b**) emission spectrum, and (**c**) change in the rms bunch durations.

## 4. Undulator Radiation of Electron Bunches in a Waveguide in the NM Regime

Consider the radiation of electrons with the same initial parameters, which move along the same helical trajectories and in the same waveguide as those studied in Section 3, but in a combined undulator and strong guiding magnetic fields. If conditions (3) are fulfilled, the NM regime is realized, under which the Coulomb interaction of electrons leads to their mutual attraction (longitudinal bunching). At the same time, depending on the regime, the excited wave can both contribute to and prevent the bunching of particles.

For a waveguide with a diameter of 4 mm and electrons with an energy of 6 MeV and initial transverse velocity $\beta_{\perp 0} = 1/\gamma_0$, the group synchronism can be achieved for the undulator period $d = 6.3$ cm, undulator field amplitude $H_u = 0.7$ kOe, and longitudinal field $H_0 = 30$ kOe. For broadband radiation with a center frequency of about 0.4 THz, the radiation efficiency is close to only 6% (Figure 7). This value is smaller than for the cyclotron radiation where the Coulomb interaction does not affect the bunch size. The decrease in efficiency can be explained as follows [27]. At the group synchronism, the electron bunch is between the maximum of the decelerating phase of the wave and its "zero." Thus, the particles moving in front are decelerated more strongly, and the beam is compressed to the edge located near the neutral phase. Therefore, in the NM regime, when a decrease in the energy of a particle leads to an increase in its translational velocity, the generation partially prevents the bunch compression.

The intersection of dispersion characteristics and the simultaneous high- and low-frequency generation of a bunch with frequencies of 0.12 THz and 1.35 THz, which coincides with the corresponding frequencies of cyclotron radiation, is realized with an undulator period of 3.5 cm, undulator field amplitude $H_u = 1.4$ kOe, and guiding field $H_0 = 60$ kOe (Figure 8). A significantly higher radiation efficiency of

12.1% than for the cyclotron variant is achieved in this case at a high frequency due to the longitudinal NM stabilization of the bunch, while at a low frequency the efficiency is 2.4%.

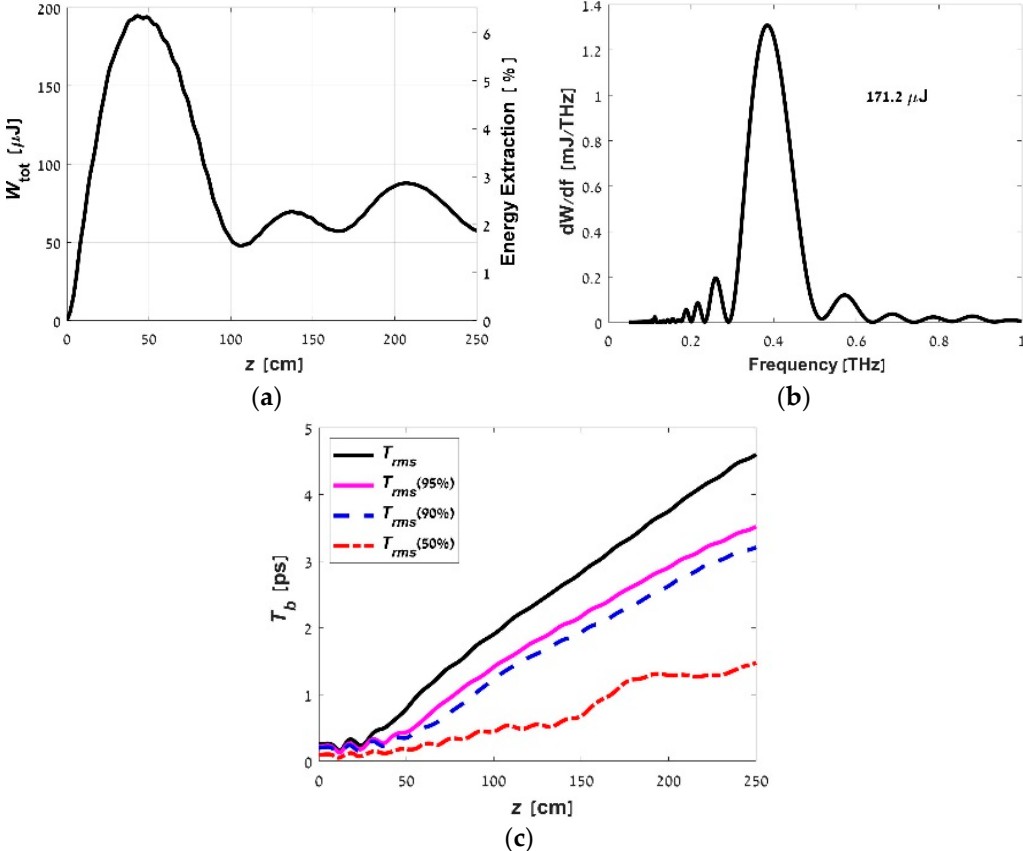

**Figure 7.** Undulator CSR of a bunch with a diameter of 1 mm, an initial duration of 0.25 ps, a charge of 0.5 nC, and a particle energy of 6 MeV in the group synchronism regime with the TE11 mode of a cylindrical waveguide with a diameter of 4 mm: (**a**) radiated energy, (**b**) emission spectrum, and (**c**) change in the rms bunch durations.

As was already noted, in this regime the low-frequency wave provides an additional stabilization of the electron bunch first in length and then in phase, i.e., the effects of the Coulomb field and the field of low-frequency radiation are unidirectional and consistent. Due to a large slippage of the low-frequency pulse relative to the electron bunch, the center of the bunch quickly shifts from the maximum of the decelerating phase towards the neutral phase of the wave, while the electron speed at the edges of the bunch increases more slowly than at the center, which leads to the formation of a peak in the charge density. Coulomb fields provide an additional stabilization: The remaining electrons from the "tail" of the bunch are pulled to the core, and the resulting dense ultrashort bunch is maintained by its own Coulomb field.

In a wider operating waveguide, simultaneous excitation of low- and high-frequency waves occurs in many waveguide modes (Figure 8) [28]. With similar parameters of the electron bunch in a wide waveguide, a short pulse is excited with a lower intensity than in a narrow waveguide. However, due to the longitudinal NM stabilization and additional bunching in the radiated-wave field, the bunch core remains compact at a much greater length (cf. Figure 6). In this case, an electron bunch of small transverse size propagating along the waveguide axis mainly excites a few dominant modes with the azimuthal index m = 1 at the fundamental undulator harmonic s = 1 with the highest energy emitted in the mode whose group velocity is the closest to the longitudinal velocity of the particles (TE$_{13}$ mode in Figure 9).

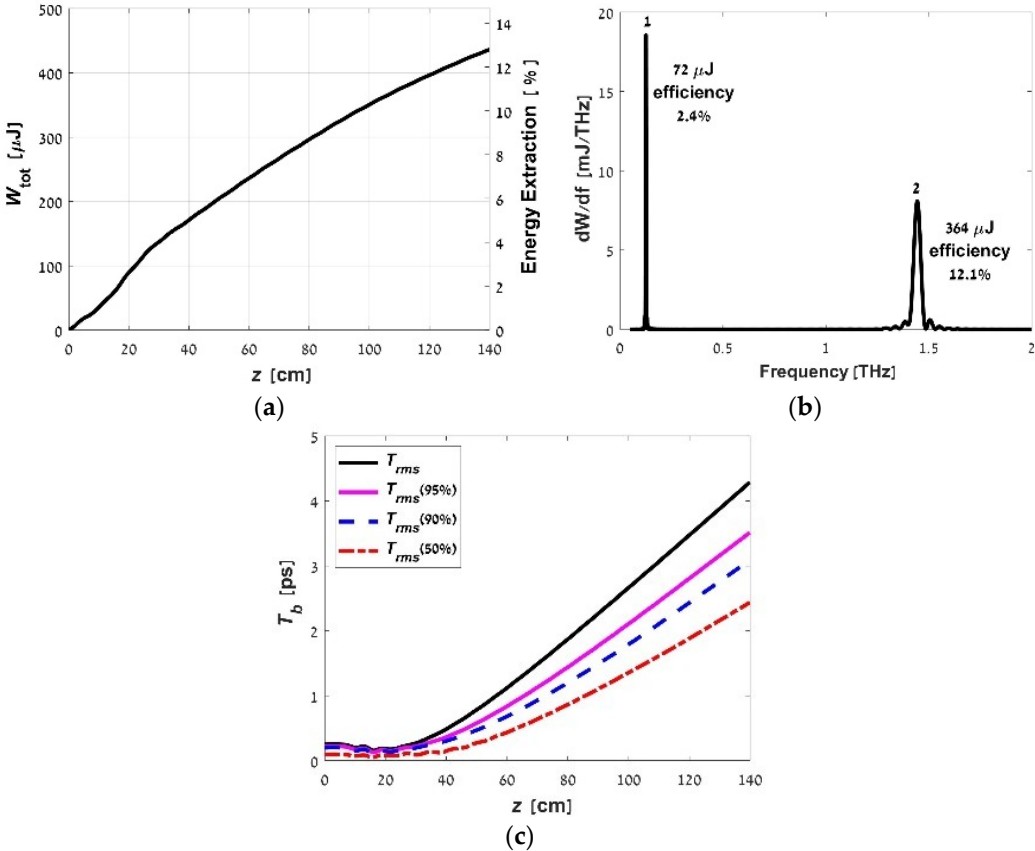

**Figure 8.** Undulator CSR of a bunch with a diameter of 1 mm, an initial duration of 0.25 ps, a charge of 0.5 nC, and a particle energy of 6 MeV in the regime of simultaneous excitation of low- and high-frequency waves: (**a**) radiated energy, (**b**) emission spectrum, and (**c**) change in the rms bunch durations.

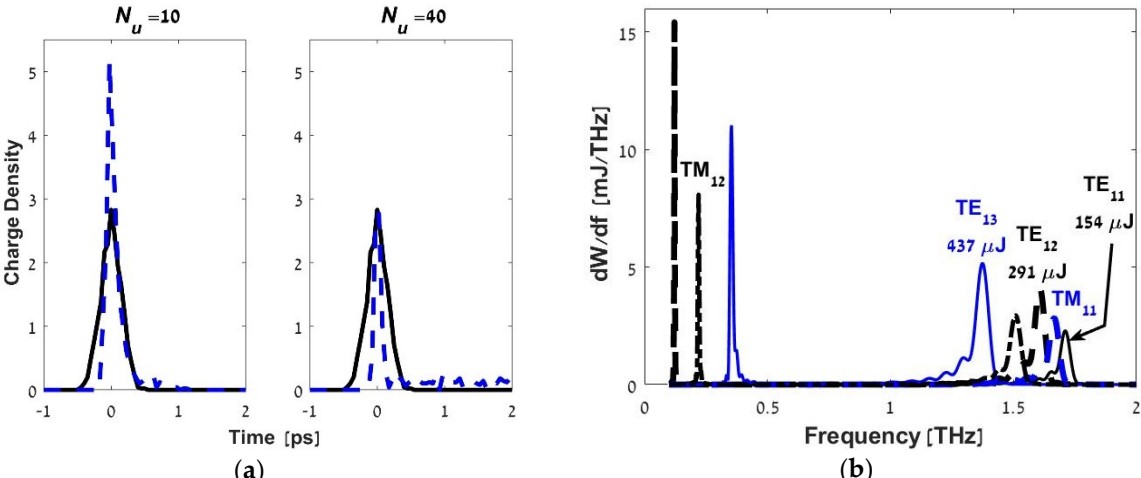

**Figure 9.** Multimode undulator CSR of a bunch with a diameter of 1 mm, an initial duration of 0.3 ps, a charge of 1 nC, and a particle energy of 6 MeV in the regime of simultaneous excitation of low- and high-frequency waves in a waveguide with a diameter of 10 mm: (**a**) evolution of the particle density from the initial Gaussian distribution (black curves) to distributions after 10 and 40 undulator periods (blue lines) and (**b**) emission spectrum for the undulator period d= 2.5 cm, undulator field amplitude $H_u = 2$ **kOe**, and longitudinal field $H_0 = 75$ **kOe**.

## 5. Conclusions

Theoretical results obtained in [13–15] have demonstrated the possibility of effective coherent spontaneous undulator and cyclotron terahertz radiation from short and dense electron bunches in undulators with a super-resonance guiding magnetic field and in a uniform magnetic field. This paper presents an analytical and numerical comparison of these two mechanisms for cases in which the particles of the bunches move along the same helical trajectories in free space or in the same regular waveguides, but due to the different dynamic properties of the particles interact differently with the fields of the space charge and the radiated waves and, therefore, provide different radiation efficiencies.

As in the case of superradiance of extended bunches [22,23], the considered cyclotron radiation of short bunches, whose longitudinal sizes are smaller than the radiated wavelength, in a regular waveguide is most effective in the group synchronism of waves and particles. It was also shown in [13] that an increase in the length of an initially short bunch due to the space charge forces does not remove the particles from resonance with the emitted waves, but leads to the transformation of the bunch into an extended, but still coherently radiating spatial structure. According to the results of numerical simulations, the bunches with parameters typical for modern laser-driven photo injectors can radiate under such conditions in sub-terahertz and terahertz ranges with an energy efficiency larger than 10% at lengths of the order of 1 m. The radiation of bunches in the group synchronism regime has a fairly wide spectrum with a relatively low center frequency, but it is weakly sensitive to the spread of the particle parameters. Much higher radiation frequencies can be generated for higher magnetic fields when the dispersion characteristics of the wave and particles intersect—in a regime close to the so-called cyclotron autoresonance. It should be noted that in this case, due to the partial compensation for the mechanisms of the inertial bunching of particles and a relatively rapid slippage of the emitted electromagnetic pulse relative to the electron bunch, coupling of electrons with the radiated waves is much weaker and bunches with the same parameters radiate with a lower efficiency than in the group synchronism. However, with a larger charge, much narrower and higher-frequency generation is possible in the autoresonance regime with an efficiency of a few percent, but with approximately the same energy of the radiated pulse as with the group synchronism. By varying the magnetic field in the radiation section of the cyclotron source with fixed parameters of the bunches it is possible to control the radiation frequency over a wide range.

In a combined helical undulator and super-resonance uniform longitudinal field, the particles of the bunches can move along the same helical trajectories as in the cyclotron case, but with an oscillation period equal to the undulator period. Under such conditions, the bunches moving in the same regular waveguide emit the same frequencies as in the cyclotron version. At the same time, the dynamics of bunches and the efficiency of their radiation in the cyclotron and undulator variants differ significantly. In the group synchronism regime in an undulator with a strong guiding field, the emitted electromagnetic pulse acts on particles in the opposite direction, degrading their bunching in the space charge field due to the NM effect, which leads to a decrease in the radiation efficiency by several times compared to a similar regime with cyclotron radiation. However, for a significantly higher-frequency radiation, these effects act in one direction. In addition, a more efficient bunching of particles and longitudinal stabilization of their distribution are also facilitated by a low-frequency wave that is excited by the bunch simultaneously with a high-frequency wave. As a result, in the NM regime, high-frequency, and relatively narrow-band electromagnetic pulses can be obtained with an efficiency of more than 10%.

In a waveguide of a sufficiently small diameter, a bunch moving near the axis of the waveguide radiates mainly at the fundamental harmonic of oscillations $s = 1$ into the lowest $TE_{11}$ mode. It is important that with the initial longitudinal size of a bunch of the order of one-half of the corresponding wavelength even at relatively large transverse particle velocities, such a bunch weakly radiates at high harmonics $s > 1$ into the modes with azimuthal indices $m = s$. In a wider waveguide, the same bunch also radiates with high efficiency basically at the fundamental harmonic, but in a series of dominant

modes with azimuthal indices $m = 1$. In this case, the highest radiation efficiency is achieved for the high-frequency mode with the lowest group velocity.

The study shows that both cyclotron and undulator radiation mechanisms make it possible to use dense sub-picosecond electron bunches to efficiently obtain powerful pulses of coherent spontaneous sub-terahertz and terahertz radiation with the ability to control radiation characteristics over a wide range of parameters.

**Author Contributions:** Conceptualization V.B., Y.O., and A.S.; investigation, V.B., Y.L., and Y.O.; writing—original draft, V.B. and Y.O.

**Funding:** This work was supported by the Israeli Ministry of Science, Technology and Space and by Russian Foundation for Fundamental Research, project 18-32-00351.

**Conflicts of Interest:** The authors declare no conflict of interest.

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
