# Peer review of "Capabilities of Terahertz Cyclotron and Undulator Radiation from Short Ultrarelativistic Electron Bunches"

_instruments, doi:10.3390/instruments3040055_

Round 1

Reviewer 1 Report

See file uploaded.

Author Response

We really appreciate your thorough work in evaluating our article. In our answer and changes made to the text, we tried to discuss and take into account all your comments and suggestions for improving the text. Please see the attachment

Reviewer 2 Report

This fundamental discussion of coherent spontaneous cyclotron and undulator radiation of short dense electron bunches is relevant to the growing capability of laser-driven photo-injectors to create short dense electron bunches. The manuscript is clearly written and it should be published.

Before publication the text should receive minor revision.

Lines 177-180: The caption for Figure 4 appears to contain the original text from the caption style guidance template.  The correct text for this caption should be inserted here.

Note also there is a relatively  minor style issue because the caption style guidance recommends that the letters (a), (b), (c) etc should be placed before the corresponding descriptive text in the captions, whereas at present in the captions in this manuscript the letters are placed after the corresponding text. 

Author Response

We are grateful to the Referee for his review.

Lines 177-180: The caption for Figure 4 appears to contain the original text from the caption style guidance template.  The correct text for this caption should be inserted here.

This fixed (176-179)

Figure 4. (a) The cross section of a cylindrical waveguide in which an electron bunch moves along helical trajectories; (b) Dispersion characteristic for the cases of the electron velocity is close to the group velocity of the radiated wave (“grazing”, G), and for the two waves at low (L) and at high (H) frequencies generation regime.

Note also there is a relatively  minor style issue because the caption style guidance recommends that the letters (a), (b), (c) etc should be placed before the corresponding descriptive text in the captions, whereas at present in the captions in this manuscript the letters are placed after the corresponding text.

 We took into account this notice and placed (a), (b), (c) etc before the corresponding descriptive text.

Reviewer 3 Report

The paper is devoted to coherent undulator radiation of a short electron bunch. This issue became topical after recent development of RF electron guns with photocathodes and short-pulse lasers. Such electron accelerators provide sub-picosecond electron bunches with low emittances and energy spread and high enough electric charge. The paper considers different operation modes of terahertz radiator with strong longitudinal field. The comparison of cyclotron and undulator modes is of great practical interest. Particular remarks are listed below.

Remarks

P. 3, Eq. (6). Seems, t~ is t? P. 4 and 5, captions of Figs. 2 and 3. Is “Coherent spontaneous cyclotron/undulator radiation of a moving plane” just “Cyclotron/undulator radiation of a moving plane”? P. 5, line 176. Sentence is not finished. P. 5, lines 177-180. Caption of Fig. 4 looks strange. P. 6, line 197. Does “good coincidence” mean “good agreement”? P. 6, line 232. The equation has to be derived, as the region of its applicability is not clear. For example, it does not work for TM waves. Moreover, symbol “Δ” is lost at the beginning.

I recommend to publish the paper after minor corrections.

Author Response

We are grateful to the Referee for his review. In the revised version of the paper, we tried to meet all Referee’s recommendations. Please see the attachment.

Round 2

Reviewer 1 Report

Even if the changes in the paper have been only superficial and the Authors didn't go deeply in answering to my comments doing only a cosmetic revision,

I consider now the paper worth of publication on Instruments.